# Organic Semiconductor Micro/Nanocrystals for Laser Applications

**DOI:** 10.3390/molecules26040958

**Published:** 2021-02-11

**Authors:** Javier Álvarez-Conde, Eva M. García-Frutos, Juan Cabanillas-Gonzalez

**Affiliations:** 1Instituto de Ciencia de Materiales de Madrid (ICMM), CSIC, Cantoblanco, E-28049 Madrid, Spain; javier.alvarez@imdea.org; 2IMDEA Nanociencia, Calle Faraday 9, Ciudad Universitaria de Cantoblanco, E-28049 Madrid, Spain

**Keywords:** organic molecules, single crystals, molecular packing, lasers, optical resonators

## Abstract

Organic semiconductor micro/nanocrystals (OSMCs) have attracted great attention due to their numerous advantages such us free grain boundaries, minimal defects and traps, molecular diversity, low cost, flexibility and solution processability. Due to all these characteristics, they are strong candidates for the next generation of electronic and optoelectronic devices. In this review, we present a comprehensive overview of these OSMCs, discussing molecular packing, the methods to control crystallization and their applications to the area of organic solid-state lasers. Special emphasis is given to OSMC lasers which self-assemble into geometrically defined optical resonators owing to their attractive prospects for tuning/control of light emission properties through geometrical resonator design. The most recent developments together with novel strategies for light emission tuning and effective light extraction are presented.

## 1. Introduction

Since the turn of century, organic semiconductor micro/nanocrystals (OSMCs) [1,2,3,4,5,6,7] have attracted continuous attention as a promising research topic with promising electronic and optoelectronic applications, including organic field-effect transistors (OFETs) [8,9,10,11], organic photovoltaics (OPVs), organic light-emitting diodes (OLEDs) [12,13], photodetectors (PDs) [14,15], and lasers [16,17,18,19,20]. A huge amount of OSMCs have been developed, with several assets over their inorganic counterparts, such us a plethora of molecular structures with diverse properties, low-cost device fabrication, compatibility with stretchy, flexible, and lightweight moldable substrates, etc. [21]. Consequently, the low thickness, light weight, foldability, and stretchability of OSMCs make them suitable, for instance, for flexible or miniaturized organic electronics and optoelectronic applications. On the other hand, OSMCs offer the possibility to be shaped into diverse molecular assemblies, which are finely tuned by crystal engineering [22]. Single crystals of high quality offer long exciton diffusion length and long lifetimes which are attractive for light-to-energy conversion [23,24]. Finally, they present large compatibility with nonexpensive solution-processed methods which can be easily scaled-up for device fabrication [25]. Different techniques such as spin-coating, drop-casting or ink-jet printing can be used with organic single crystals for the development of electronic devices. The high purity and low density of defects in organic single crystals is crucial for the development of high-performance devices and circuits. To optimize device performance, it is necessary to avoid grain boundaries, defects, impurities, and dislocations.

OSMCs can be prepared into different micro/nanostructures with controlled fabrication procedures, enabling for different properties, depending on the nucleation, molecular packing, and assembly [22]. The formation of OSMCs is determined by regular and stretched intermolecular packing among neighboring molecules with moderately weak noncovalent bonds, such as Van der Waals forces, hydrogen bonds, and the π-π interactions. This affords different packing modes that are decisive in the formation of organic micro/nanocrystals which highly influence their optoelectronic properties. These intermolecular interactions are also easily influenced by the external conditions like the light, solvents and the temperature.

The study of both crystal growth and engineering for the preparation of high-quality OSMCs has been widely addressed. An in-depth analysis of the crystal engineering allows getting a variety of different micro/nanocrystals with various physical properties. Herein, one of the most important aspects is the chemical versatility and modular nature of organic materials, allowing for modulation and change in the intermolecular interactions through subtle changes in the molecular structure. Moreover, physical properties such us melting point, sublimation temperature, or solubility are also important aspects to consider for OSMC growth. Thus, many organic semiconductors with diverse molecular structures have been synthesized and described.

Therefore, taking into account the dimensionality and the shape of the microstructure/nanostructure, a control of the different properties can be achieved. OSMCs can be prepared into 1D wires, tubes, 2D-sheets, belts or discs, affording different crystals morphologies [5,7]. 

Meanwhile, OSMCs development has not only restricted to achieving randomly dispersed organic crystals, but also there have been huge efforts in developing methods for alignment and patterning of OSMCs into ordered arrays, with the goal of obtaining much better device performance [4,26].

There is an important relation between the molecular structures, packing modes, crystal morphologies, and optoelectronic properties (Scheme 1), achieving the final and complex property of the material. 

In this review, we firstly introduce the importance of molecular packing in the crystals. The different noncovalent intermolecular interactions such as hydrogen bonding, π-π stacking, van der Walls forces amongst others, govern the packing arrangement of the molecules, aside from electronic and optoelectronic properties. Subsequently, we discuss the different large number of organic crystal growth techniques for controlling the crystallization of organic semiconductors to get OSMCs. Finally, we discuss the main developments in the field of organic crystal lasers, describing briefly the photophysics and figures-of-merit of OSMCs in terms of optical gain properties, the types of crystalline optical resonators more commonly reported and recent relevant examples of laser cavities based on the different types of crystal resonators.

## 2. Molecular Packing in Crystals

In organic semiconductors absorption and emission of light implicates an electronic transfer between the highest occupied molecular orbital (HOMO) and the lowest unoccupied molecular orbital (LUMO) which forms and removes an exciton. Optical properties of these semiconductors differ in solution and in the solid state (Figure 1a). In solution organic molecules are surrounded by solvent molecules which implies no disruption in exciton formation, whereas in solid state molecules of the semiconductor are close to each other involving a direct overlap of the molecular orbitals (MOs) of neighboring molecules, creating excitonic couplings. One example is the interaction of transition dipole densities, affecting the optical and electrical properties of the material [27,28,29,30,31].

As mentioned before semiconductor molecules aggregate by noncovalent, weak intermolecular interactions as hydrogen bonds, π-π stacking, Van der Walls forces amongst others. These forces are the ones that govern the molecular packing the molecule undergoes. There are four typical packings [32] (Figure 1b,c), the first one consisting of two adjacent molecules that are arranged completely face to face, called an ideal π stacking (Figure 1b I), and stands to an H-aggregate in relation to their optical properties. This arrangement gives the largest intermolecular overlap which leads to a decline in the optical properties due to the strong π-π overlap; on the other hand, it is the packing that allows for efficient charge transfer and high mobility alongside the stacking direction. To achieve this packing is hindered by the high electrostatic repulsion of the neighboring molecules [33]. Generally, a face-to-face arrangement involves a slight translation along the Y plane between the two molecules. This situation is referred as a pitched π-stack which is defined by the pitch (P) angle [34,35] (Figure 1b II, c I). By increasing the pitch angle by 50% or more slipping, the H-aggregation changes to a J-aggregation and the overlapping of the orbitals and splitting energies are reduced, providing better emission properties [36,37,38]. Another way of achieving this is by moving the stack on the X plane, achieving a rolled π-stack (Figure 1b III,c II), defined by a rolled (R) angle. When the π-π overlap is decreased, the exciton created localizes quickly, making the intermolecular vibrations barely participative in the emission spectra. Contrarily, when the π-π overlap increases the charge transfer (CT) increases, promoting an intermolecular separation upon electronic de-excitation which implies a loss of vibronic structure, red-shift and excimer emission features [17]. One example of the pitched and rolled arrangements is 1,4-bis(*R*-cyano-4-diphenylaminostyryl)-2,5-diphenylbenzene(CNDPASDB) [39,40,41] (Figure 2a) in which the molecules are stacked but shifted along the X or Y plane. Another arrangement we can encounter is the herringbone (Figure 1b IV,c III) in which one of the molecules is rotated along its long axis with an angle (H), getting an edge to face alignment. This packing decreases the π-π stacking as in the J-aggregates allowing better emission properties. Pentacene and 1,4-bis(4-methylstyryl)benzene (p-MSB) crystals are examples of a herringbone motif [42,43,44] (Figure 2c). The last arrangement we can encounter is the X-aggregation (Figure 1b V, c IV), when one molecule rotates around the stacking axis but retaining the molecular planes parallel with each other. It is the arrangement that in theory should give rise to the strongest fluorescence properties due to the least π-overlapping and large molecular distance and high carrier mobility, depending on the rotating angle [45,46,47]. An example of the X-aggregate takes place in perylene-3,4,9,10-tetracarboxylic tetrabutylester (PTE) depicted in Figure 2e wherein one molecule is rotated with respect of the other 70.2° [48].

Another aspect to bear in mind is the presence of intralayer molecular interactions which are much weaker than the face-to-face interactions; these interactions produce a tilting in the molecular layer which can be measured by an angle (L) between the normal of the bottom crystal plane and the molecular long axis (Figure 2). Depending on the interactions between the layers there can be a “zig-zag” disposition (Figure 2b,d). Figure 2b for instance displays 9,10-bis((*E*)-2-(pyrid-2-yl)vinyl)anthracene (BP2VA) molecules having a pitch or roll packing but with a “zig zag” arrangement between layers [49,50]. In addition, p-MSB crystal has a herringbone motif but with a “zig zag” arrangement between layers [51].

The different H-, J- and X-aggregate arrangements give raise to changes in the electrical and optical properties depending on the exciton and splitting energy. (Figure 3). Focusing on the optical properties, H-aggregates lead to an absorbance blue shift (hypsochromic) respect to solution concomitant with a low radiative constant (K_r_), whereas J-aggregates absorbance exhibit a red shift (bathochromic) and a high K_r_ [36,52]. In the case of X-aggregates generally the absorbance in solution and in the aggregate itself is similar. As aforementioned H-aggregates often causes quenching in the solid state due to the strong π-overlap, but in the case of J-aggregate or herringbone packing the π-overlap decreases, so the optical properties improve. A strategy to achieve good mobility and emission is to join simultaneously these two, J-aggregate and a herringbone packing, or the use of the X-aggregates which reduces the π-overlap but maintains the planarity [53,54,55,56].

## 3. The Growth Techniques for the Preparation of Organic Semiconductor Micro-/Nanocrystals

The methods used nowadays for inorganic crystal formation are not adequate for organic single molecules due to the harsh conditions used such as high pressure and temperature, hard reaction conditions and numerous solvents used [57].

The preparation methods of OSMCs can be classified into three categories: solution, melting and vapor processing. Solution processing is often used for nonthermally stable materials whereas melting and vapor processing can be used for materials with high thermal stability and low solubility[2].

A wide variety of different techniques for controlling the crystallization of OSMCs have been developed during the last decades [2,3]. A precise control of the crystallization process is key to achieve high quality crystals, therefore the growth method and conditions are essential to the morphologies and the molecular arrangements. The morphology and the stacking of these OSMCs depend on the different conditions in the growth methods. A control in these growth methods should afford a better crystal quality providing better device performance.

### 3.1. Solution-Processing Techniques

These are the most simple and effective approaches to grow organic crystals because most of organic molecules are soluble in a multitude of organic solvents in a wide range of temperatures and pressures. The concentration in solution increases upon solvent evaporation, reaching a point of saturation where molecules self-assemble creating complex micro/nanocrystalline structures [58,59,60].

#### 3.1.1. Drop-Casting

The drop-casting method is the most efficient approach to grow OSMCs by self-assembly of organic molecules. The self-assembly process depends on the intermolecular interactions between solvent molecules, organic molecules and organic-solvent molecules. The growth of the crystals with this method requires control of different conditions such as solvent, concentration, atmosphere and temperature. The growth condition is key to optimize the crystal quality. The method consists of dropping a volume of organic semiconductor solution onto a substrate and let the solvent evaporate for several hours or days. Precise control of concentration, temperature, atmosphere and substrate surface enables for the formation of high-quality crystals [61,62,63]. One example is 9,10-bis(phenylethynyl)anthracene (BPEA), which led to different crystal phases depending on the solvent used (chlorobenzene or dichloromethane) because of the evaporation velocity, which is determined by the interaction between the molecule and the solvent, and whether it is an open system or a quasi-closed system [64]. Another example is diphenylfluorenone (DPFO): adding a solution of this molecule in THF to a substrate leads to microfibers, whereas adding more solution on top caused microfiber redissolution and subsequent formation of microplates upon solvent evaporation (Figure 4a–c)[65].

#### 3.1.2. Dip-Coating

This technique consists of pulling out a substrate that is immersed in a solution of the organic molecule. This technique allows obtaining organized pattern crystals. Firstly, molecules crystallize in the substrate due to solvent evaporation. Owing to the concentration gradient and capillary forces, more molecules from the solution will move to the contact line depositing more material, crystallizing opposite to the pulling direction. The main parameter to control is the dip-coating speed which strongly influences the morphologies of the crystals formed. Nanoribbon arrays were obtained for instance from a solution of BPEA and triisopropylsilyethynyl pentacene (TIPS-PEN) with this method. Applying pulling speeds higher than 80 µm s^−1^ led to individual nanoribbons whereas lowering the speed below 30 µm s^−1^ led to a conglomerate of nanoribbons (Figure 4d–f) [4,26]. 

#### 3.1.3. Solvent Exchange

A commonly used method consists of making the solution saturated or hypersaturated by adding an antisolvent. Through diffusion of the solvent and antisolvent the molecule precipitates and self-assembles [66]. In order to implement this method, a few conditions have to be fulfilled: (i) the organic molecule must be soluble in one solvent and insoluble in the other; (ii) both solvents ought to be miscible with one another; and (iii) both solvents should have different densities in order form an interface and avoid rapid mixing which would lead to fast nucleation [67,68,69,70]. An example of this process is depicted in Figure 5a. This method led to C_60_ crystals with different plate or rod morphologies by changing the solvents and solvent ratios of CCl_4_, m-xylene and isopropanol [71]. 

Another interesting example stands for the use of this method to obtain micro-rings; this technique uses the solvent exchange method along with the interfacial tension (Figure 5b). Micro-rings are achieved by adding a droplet of a solution of 1,5-diphenyl-1,4-penta-dien-3-one (DPPDO) (a flexible compound) in ethanol/water into a substrate, the ethanol evaporation triggers precipitation of the compound on the water droplet. Nucleation starts preferentially at the drop edge, whereas the water tension and weak intermolecular interactions enable the wires to bend into micro-rings. In addition, if the concentration is increased to 10 mmol L^−1^, microtiles are obtained [72].

#### 3.1.4. Solvent Vapor Diffusion (SVD)

When avoidance of solvent mixing is an obstacle or solvents have same densities, solvent-vapor diffusion (SVD) can be used. It is a variant of the solvent exchange method with the difference that the antisolvent is placed outside rather than inside the solution. Slow evaporation of the antisolvent leads to its gradual diffusion inside solutions to mix with the solvent, provoking molecular precipitation and self-assembly. This method reduces the mixing speed of the solvents being able to achieve higher and better quality crystals [73,74,75]. High quality 2,2′,7,7′-tetrakis(*N*,*N*-di-*p*-methoxyphenyl-amine) 9,9′-spirobifluorene (spiro-OMeTAD) crystals were developed with this method (Figure 6) [75]. As shown in Figure 6a, the inner vial contains the spiro-OMeTAD solution in DMSO at a concentration of 1 mg/mL, whereby the outer contains the nondissolving methanol. When the diffusion of methanol vapor proceeds slowly to the inner vial at room temperature, it provokes a sustained reduction of solubility of spiro-OMeTAD in the methanol-enriching solution, finishing in a supersaturation state that triggers its crystallization (Figure 6b,c).

Single crystals of diperylene bisimide were also grown through this strategy. In this case the inner vial contains the solution of the bisimide in toluene and the outer vial contains methanol as the antisolvent [76].

### 3.2. Melting Processed Crystals

The melt crystal growth methods, such as Bridgman and Stockbarger, Czochralski, or floating zone methods, are often used for growing large crystals of inorganic semiconductors, however these methods have also been used for organic single crystal growth (Figure 7) [77]. Normally, these methods are known as zone refining, zone melting or zone freezing technique. They have been less employed for growing organic crystals due to the high vapor pressure and chemical stability of organic molecules around melting temperatures.

This method, however, has some constraints: (i) molecules must have a well-defined melting point; (ii) large thermal stability at fusion temperature is required; (iii) molecules must have low chemical activity; and (iv) they must be extremely pure because this technique is highly sensitive to impurities [57]. 

Among these requirements, the main challenge concerns with the thermal stability of the organic compound. In addition, these methods require a large amount of material, and relatively expensive apparatus. A way to circumvent this problem consists of growing the crystals between two glass or quartz slides [78]. Thiophene-phenyl-pyrrole (TPP) crystals were grown by this method upon placing the material between two quartz substrates in a hot stage, obtaining flat crystals that show waveguiding properties (Figure 8a) [79].

### 3.3. Vapor Processed Crystals

As mentioned above this method is used primarily for molecules with low solubility and high thermal stability. It involves phase transitions between solid, liquid and vapor phases.

#### 3.3.1. Physical Vapor Transport (PVT)

The most common technique is physical vapor transport (PVT). First proposed by Kloc and Laudise et al. [80,81] became one of the most popular methods for growing organic crystals. It consists of heating the material, most of the cases under vacuum so that the boiling point of the material lowers. The sublimated material is then transported to a lower temperature zone by an inert gas where it crystallizes. By this method impurities also crystallize in front or behind the crystallization zone. The control parameters in this technique are carrier gas flow, temperature gradient, and vacuum level (Figure 8b) [82,83,84]. Doped crystals of BSB-Me with tetracene and pentacene were grown with PVT using just the doped powder, to obtain crystals thicknesses of less than 400 nm and length of several millimetres (Figure 8c) [85].

#### 3.3.2. Microspacing Sublimation

Crystals grown from PVT are of high quality, but the growth requires high vacuum environment, inert flowing gas, high control of the temperature and expensive apparatus. In this sense Xutang Tao et al. reported a new method to obtain organic crystals through sublimation based on microspacing distance, which is low cost and requires less parameter control (Figure 9) [86]. It consists of heating the organic molecule in a hot stage deposited in a substrate until sublimation to the upper substrate, separated around 400 µm. The evaporated material condenses in droplets on the upper substrate, from which crystals grow. They obtained high quality crystals of anthracene, perylene, pentacene, pyrene among others with sizes from 10 to 50 µm [87,88].

## 4. Organic Solid Lasers 

A laser consists of an optical gain medium located in an optical cavity providing optical feedback in one, two, or three directions. Light is generated inside the medium by electrical (electrically-pumped lasers) or optical (optically-pumped lasers) stimuli and amplified by stimulated emission. Organic π-conjugated materials exhibit very interesting features as active media in laser devices. They show high room-temperature photoluminescence quantum efficiencies (PLQE) [89,90] which translates into large stimulated emission (SE) cross-section values and their photoluminescence (PL) spectra can be tuned through chemical functionalization [91]. Moreover, organic lasers can be processed by cost-effective solution-based methods [92], they have capabilities to confine and guide light due to their elevated refractive indexes [93] and they possess unique mechanical properties (flexibility and light weight) leading to new potential market niches for organic laser devices. From a more intrinsic point of view, organic semiconductors are potential four-level laser systems, (see scheme in Figure 10a). In this configuration excited states generated by an electrical or optical perturbation relax efficiently to the lowest excited state. Depending on the nature of the organic molecule such state can have a local excited [94,95], or intramolecular charge transfer character [96]. In all cases, fast internal conversion and vibrational cooling leads to fast pumping of the lowest excited state from which, radiative emission to the upper vibrational levels of the ground state occurs. Given that these upper vibrational levels are empty at room temperature, stimulated emission takes place without competition with resonant ground state absorption. This lack of competition is what endorses conjugated molecules with unique features for optical gain, thus guaranteeing laser action at low optical pumping thresholds. Another interesting feature relies on the fact that emission is displaced from the absorption spectral region by the so-called Stokes shift, minimizing re-absorption. Furthermore, spectral displacement can be achieved through the realization of co-crystals based on donor-acceptor moieties which also enable to span the luminescence across the visible [97,98]. A myriad of organic crystals are found to combine high PLQE and outstanding charge transport [85,99,100,101], paving the way for the realization of electrically-pumped lasers, the first demonstration being recently reported in 2019 [102].

A basic example of an organic laser cavity is composed by and organic semiconductor film located between two metallic or dielectric mirrors or gratings, either in external [103] or in an integrated microcavity geometry [104]. Other types of organic laser geometries are mirror-free surface-emitting distributed feedback (DFBs) cavities [105,106] where a grating is imprinted on (or formed by refractive index variation of) the gain medium, producing feedback in one direction. In both cases waveguiding by total internal reflection due to the difference in refractive indices at the organic layer boundaries is required to confine the light in the direction perpendicular to the layer. 

Organic crystals provide the possibility to merge optical gain and optical feedback due to the presence of well-defined crystal faces enabling optical confinement through total internal reflection inside the crystal cavity. Despite the vast number of reports on optical gain from organic crystal lasers, the majority of them refer to processes which are not supported by cavity geometrical resonances, (e.g., random lasing or amplified spontaneous emission (ASE)). ASE does not require optical feedback because light amplification takes place by a single pass along the optical gain medium. ASE is typical of organic waveguides (slabs, 1D planar waveguides or optical fibers) [107,108,109,110,111,112] but can also be supported by organic crystals [68,113,114]. The ASE output is constituted by a spectrally broader emission linewidth (~10 nm) corresponding to the amplified waveguided mode along the crystal. Random lasing is instead associated to multiple scattering effects given raise to closed loops and optical feedback. This process is often observed in crystals which present refractive index inhomogeneities including crystal dislocations, different morphology areas or impurities. These inhomogeneities lead to multiple scattering and formation of resonances via random walk. They often display multiple lasing modes and their emission cannot be controlled though crystal design owing to their disorder nature [115,116,117].

Hereafter, we will address organic crystal lasers constituted by optical microresonators formed by boundaries of the organic gain medium itself and whose resonances are determined by the microresonator geometry and the refractive index of the medium. Different organic crystal microresonators with defined geometries have been reported, ranging from wires [118,119], fibers [120], rings [121,122], polygonal cavities [123,124], slab crystals [125] or disks [126].

### 4.1. 1D Fabry-Perot Resonators

Fabry–Perot (FP) cavities are typical of one-dimensional microstructures such as wires, fibres and hollow fibres, where photons undergo total internal reflections at the cavity walls and bounce back and forth at the cavity facets, leading to a periodical interference pattern (Figure 10b). Light confinement becomes observable when the diameter of the cavity approaches the wavelength of the light. Dimensions below the wavelength give raise of strong diffraction effects lowering considerably the confinement efficiency, which is given by the fractional mode power within the core.
(1)$$\eta =1-{\left(2.405\exp\left[-\frac{1}{V}\right]\right)}^{2}V^{-3}$$
where V = πd/λ (n^2^ − n^2^_0_)^0.5^, d is the wire diameter, and n and n_0_ stand for the refractive index of the wire and surrounding medium (air). For n ~ 1.7, characteristic of an organic medium, and λ = 460 nm, a confinement efficiency of > 85% is expected when *r* amounts to 300 nm [127]. Spontaneous emission gives rise to a given spectral distribution *I(**λ**),* the light outcoupled from the Fabry-Perot (FP) resonator being given by:(2)$$I_{t}=I\left(\lambda \right)\frac{{\left(1-R\right)}^{2}}{{\left(1-R\right)}^{2}+4Rsin^{2}\left(\frac{2\pi }{\lambda }L\right)}$$
where *R* and *L* stand here for the reflectance at the cavity facets and the length of the cavity respectively. The light outcoming the FP displays characteristic periodical resonances arising from longitudinal modes, spectrally separated by Δν~c/2 nL. The number of longitudinal modes that a certain cavity can support is given by the ratio Δν_sp_/Δν, where Δν_sp_ stands for the spectral bandwidth of spontaneous emission.

### 4.2. Whispering Gallery Mode Resonators

Whispering gallery mode (WGM) resonances are a tangible phenomenon when referred to sound waves. Many of us has surely experienced before the capability of curved walls from arches or domes to propagate sounds as weak as whispers. Like sound, electromagnetic waves experience an analogous effect. In curved surfaces based on highly dense medium, light experiences multiple total internal reflections at the medium-air interface leading to closed loops where light interferes constructively. The result is a standing wave pattern distributed along the curved surface (Figure 10c) [122]. WGMs are present in spheric and hemispheric cavities and cavities with circular surfaces like cylinders, or rings. Cavities of polygonal shape can also support WGMs. The number of loops that photons undergo before leaving the cavity is given by the quality factor (Q). This magnitude is defined as: (3)$$Q=\frac{\lambda }{\Delta \lambda }$$
and expresses the capability of the cavity to trap light. Q factors in WGM resonators can be as high as 10^11^ although in organic WGM resonators values typically range between 10^3^ and 10^4^ [128]. A direct consequence of a high Q-factor is the concentration of high optical power in the resonator giving raise to strong-light matter interactions. WGM resonators can be applied to the development of lasers with low pumping thresholds and very narrow linewidth. In these cavities the spectrum of the confined light experiences a rippling ascribed to the multiple confined modes. The spacing between these resonances is inversely proportional to the cavity diameter within the geometrical approximation:(4)$$\Delta \lambda =\frac{\lambda^{2}}{\left(n-\lambda \frac{dn}{d\lambda }\right)L}$$

A direct implication of EQ.3 and EQ.4 is that the Q factor increases with increasing the cavity diameter. An exact treatment requires the solution of the spherical or cylindrical vectorial electromagnetic boundary problem, i.e., Mie theory or the equivalent for cylindrical structures. Transverse electric (TE) and transverse magnetic (TM) solutions result from these calculations. Whereas microresonators generally support various lasing modes within the spectral range of the gain material, it is possible to achieve single mode lasing by coupling two such resonators of different sizes [129]. An interesting feature of WGM cavities is that they have extremely high Q-factors, meaning that photons in these structures undergo many round trips, which makes them interesting for low threshold lasing [130] and highly sensitive transduction of physical or chemical perturbations [131]. Consequently, WGM cavities have received an ever-increasing interest for biosensing applications [132], imaging in biological media [133], and physical sensing [131].

### 4.3. Lasers Based on 1D Fabry-Perot Resonators

Organic molecules arranged in one-dimensional structures supported by intermolecular interactions, such as hydrogen bonding, π-π or halogenated bonds display unique photonic properties in terms of light transport and optical amplification. Structures with highly defined flat end faces can behave as efficient FP cavities producing the required feedback to achieve laser action (Figure 11a,b). Crystalline one-dimensional wires are one example [134,135,136]. The length and width of the wire as well as the molecular orientation determine whether the FP cavity is constituted along the wire, or between the lateral faces of the wire. A clear indication of the resonance direction comes from analysis of wires with different lengths (*L*) and the assessment of the inversely proportional relation with mode spacing (Δ*λ*) given by EQ.4 (Figure 11c,d). Crystalline wires with longitudinal FP modes (i.e., those yielding from oscillations between the wire end tips) were for instance reported by Wang et al. based on a methoxyphenyl-hydroxynaphthalen (DMHP) derivative [119]. Single crystals of this compound obtained with the solvent-exchange method were composed of molecules arranged in orthorhombic structure with a unit cell composed of four symmetrically equivalent molecules, growing in one direction. The resulting wires had lengths ranging from 5 to 100 μm and widths from 50 to 250 nm with very smooth lateral faces of few nanometers roughness which displayed red emission and PLQEs of 32%. The PL spectrum was decorated by the characteristic periodically displaced modes attributed to FP cavities and the separation between the modes followed a linear relation with *1/L*. Multimode lasing centered at 720 nm was observed upon pumping with fluences above the 1.4 μJ cm^−2^ lasing threshold. 

Wires with large widths can in some cases support FP cavities defined between the lateral faces. An example is the work by Fu and co-workers on 1,4-dimethoxy-2,5-di[4′-(methylthio)styryl]benzene (TDSB) [118]. Single crystals of TDSB arrange into H-aggregates forming a monoclinic structure. The resulting wires have a characteristic rectangular shape of 0.5–2 m width and variable lengths up to 100 μm. Herein, the tight molecular co-facial packing and strong electron-phonon coupling in the TDSB crystals enhance the oscillator strength of the H-aggregate allowed transition leading to a PLQE of 81%. Differing from the previous example, the cavity mode spacing arising from these wires was independent of the wire length but followed a linear inverse dependence with the cavity width, confirming that the FP cavity is built-in between the wire lateral faces. Restricting the dimensions of the same wires in the transversal direction can be an effective tool to design the resonant cavity geometry. This can be achieved for instance with capillary bridge lithography [123,137]. This method exploits de-wetting of a liquid film deposited on a substrate by placing the solution in contact with a template of periodically arranged ribs with their surface modified with heptadecafluorodecyltrimethoxysilane (Figure 12a–c). Capillary forces drive the solution underneath the pillars where slow solvent evaporation triggers molecular nucleation leading to crystal patterns which follow the rib shape. Capillary bridge lithography was employed to obtain TDSB wires of same crystal structure and rectangular section as those obtained through self-assembling in solution but of lesser width (0.5 × 0.5 μm) [123]. Detailed analysis of the PL spectra and mode spacing of wires with different length confirmed that the resonant modes arose from longitudinal optical oscillations between the wire tips. 

FP wires typically exhibit multimode lasing although single mode lasing can be achieved in short length cavities where the mode spectral separation Δ*λ* is larger than the optical gain spectral bandwidth of the organic crystal. Liao et al. reported multimode lasing in microbelt-shaped crystals of 1,4-dimethoxy-2,5-di[4′-(cyano)styryl]benzene, an organic compound which displays J-aggregation [138]. Microbelts supported FP resonant modes between the lateral faces and displayed different mode spacing dependent on the microbelt width. Some of these crystals with the smallest widths (625 nm) gave rise to single mode lasing at 531 nm. In principle, single mode lasing is particularly interesting because it lacks from competition between multiple modes, which a priori would lead to lowering of the lasing thresholds. Nevertheless, the threshold achieved in these single mode cavities were somewhat larger than similar multimode cavities, a result which could be explained by the individual characteristics of the measured microbelt. 

Regarding the laser emission characteristics of organic FP resonators, the different works report laser lines which span from 500–750 nm depending on the compound, with lasing thresholds which are typically in the 0.1–10 μJ cm^−2^ range achieved by pumping with 100–300 fs pulses at 1 kHz repetition rate, (see Table 1 and Figure 13 for a detailed description). The spectral position of the lasing lines is determined by the gain spectral bandwidth, which typically corresponds to the bandwidth of the 0–1 vibronic PL peak, and the cavity modes supported within this bandwidth. Interestingly, in some cases optical gain from 0–1 and 0–2 can coexist leading to simultaneous lasing at two spectral regions. Wang et al. demonstrated that short wires of DMHP exhibited multimode lasing centered at 660 nm whereas long wires exhibited a shift of the peaks towards 720 nm, coinciding with the 0–1 and 0–2 vibrational PL peaks of DMHP [119]. This effect was explained as due to the long DMHP absorption tail extending down to the 0–1 PL spectral region. Under these circumstances, losses by self-absorption at 660 nm become important and lasing can only be achieved through stimulating the emission from the 0–2 transition, although at expenses of a significant increase in lasing threshold. Wires of 30.7 μm length exhibited dual laser emission centered in these two spectral regions.

### 4.4. Lasers Based on Whispering Gallery Mode Resonators

Organic molecules with tendency to self-assemble along two crystal growth directions rather than one preferential direction offer the possibility to achieve 2D and 3D crystalline geometries with capability to behave as WGM optical resonators [128]. This is the case for instance of p-distyrylbenzene (DSB) or 1,4-dimethoxy-2,5-di[4′-dimethylamino-styryl]benzene (DASB). DSB self-assemble into lamellar herringbone aggregates leading to an orthorhombic lattice with perfect square symmetry, depicting sharp spots in the selective area diffraction pattern characteristic of a single crystal [124]. The resulting crystals are hexagonal microdisks with edges of 1–5 μm size and thicknesses of about 350 nm (Figure 14a). These crystals are highly fluorescent with typical PLQEs of 65% arising from the large oscillator strength of the transition between the upper level in the H-band and the ground state. The PL spectrum of a single hexagonal disk appears decorated with resonant modes, their density becoming larger as the size of the hexagon increases. Information into the exact geometry of these resonances inside the hexagons is inferred from the single crystal fluorescence images and from the refractive index group estimated from mode spacing analysis, assuming three possible resonance geometries. The fluorescence image of a given microdisk (Figure 14b) depicts an alternate bright-dark edge, thus confirming that a closed loop involves only three total internal reflections (3-WGM or 3D-WGM) at alternate faces, (Figure 14c). Among these two resonance geometries, D3-WGM is more plausible based on the feasible group refractive index value estimated from EQ.4. The resulting hexagonal microdisks behave as laser cavities with lasing thresholds as low as 0.79 μJcm^-2^ and the possibility to transit from multimode to single mode lasing by shrinking the hexagon edges from 5 to 1 μm (Figure 14d). More complex octahedron microcrystals were obtained from self-assembling of DASB which showed PLQE values of 30%, and visible WGMs in the PL spectrum which were rationalized as closed loop formed by six total internal reflections at six different crystal edges [126]. In line with DSB hexagonal disks, DASB octahedrons also led to multimode lasing with the possibility of achieving single mode lasing by restricting the cavity size. The Q-factors of these crystals ranged from 1500 to 7900. The Q-factors of WGM resonators are typically larger than those from FP resonators. This is basically a consequence of the WGM resonator geometry which facilitates total internal reflections because light travels more parallel compared with close to normal incidence at the FP ends. Following this same argument, larger Q-factors are expected from WGM resonators of smaller curvatures (e.g., polygonal cavities of high order) or of larger sizes [121,139].

Control of light emission in these cavities has also been achieved by patterning [121,123] and solvent-assisted methods [139]. Growth of a OH-substituted 3-[4-(dimethylamino)phenyl]-1-(2-hy-droxyphenyl)prop-2-en-1-on (HDMAC) into 1D-wires or 2D-microdisks was achieved through self-assembling from protic or aprotic solvents respectively [139]. The molecules pack along the [010] and [002] crystal direction with the OH- groups piling along this plane. Hydrogen bonding of protic solvent molecules at the OH position interferes with [002] growth leading to 1D wires along the [010] direction. The use of aprotic solvents enables instead the formation of rectangular microdisks with WGM resonance loops defined by four reflections at each of the disk faces. TE and TM polarized modes characteristic of WGM resonators were clearly visible in the PL spectrum. Large resonators of about 55 μm round trip exhibited the largest Q-factors (~7000). The lowest laser thresholds achieved were 0.4 μJcm^−2^. 

Control of sizes and shapes in WGM resonators has also been achieved by lithography methods. Capillary bridged lithography was applied by Honbing Fu and co-workers to obtain indistinctly WGM and FP resonators [123]. They obtained micro-rings with different number of WGMs upon tuning the diameter of the pillar template which determines the ring diameter. Thus, 8, 12 and 16 μm diameter rings were achieved which led to 3, 5 and 7 optical modes in the PL spectra. Multimode lasing was achieved with the lowest threshold at 0.3 μJcm^−2^. The validity of this method to achieve small diameter WGM resonators capable to support single mode lasing remains nevertheless under question due to the low mechanical resilience of organic crystals which could develop cracks when patterned into high curvature surfaces. A different patterning method to achieve WGM resonators of controlled size was reported by Fang et al. based on reactive-ion etching (Figure 12d) [121]. A typical realization consisted of depositing either a 5,5’-bis(4-biphenylyl)-2,2’-thiophene (BP1T) or a 5,5”-bis(biphenyl-4-yl)-2,2′:5′,2”-bithiophene (BP2T) layer of nanometer roughness by PVD onto a substrate and spin-coating subsequently two layers of PVA and SU8 photoresist. The desired motifs were transferred into SU8 by UV lithography and temperature annealing leading to coated patterns of PVA and irradiated SU8 on top of the crystalline film. This method required applying baking temperatures below 70 °C to avoid organic film cracking. The interstitial film between the motifs was then removed by reactive ion etching. Finally, the PVA/SU8 capping was lift-off using an appropriate orthogonal solvent. This method enables the development of a wide range of polygonal disks with triangular, square, circular, pentagonal, hexagonal or star-shape sections and of different sizes, (Figure 12e). Although it offers an attractive way to shape the resonator, perhaps the most interesting result in practical terms is the possibility to control the resonator size, because the standard self-assembling methods give raise to a wide distribution of sizes. The best performing WGM resonators were those patterned into rings showing Q-factors of 2030 and narrower lasing modes respect to polygonal cavities. 

An important aspect of optical resonators is the development of strategies to enhance light outcoupling. In WGM resonators this has been envisaged by means of using external photonic structures (generally wires). The main challenge here is to avoid interference of the outcoupling structure on the waveguiding properties of the WGM resonator. Lv et al. demonstrated that 3-[4-(dimethylamino)phenyl]-1-(2-hydroxy-4-fluorophenyl)-2-propen-1-one (HDFMAC) self-assembles into thin micro-rings with a width smaller than the propagating wavelength and very high aspect ratio (250 nm width, 2.5 m height) [122]. This geometry leads to considerably leakage of the transversal electric (TE) mode, polarized parallel to the substrate, whilst the transverse magnetic (TM) mode locates far away from the substrate, avoiding substrate-induced propagation losses and enabling light outcoupling to an external photonic structure.

By changing the proportion of ethanol:dichloromethane in the master solution, HDFMAC was observed to assemble into micro-rings tailed with microbelts (Figure 15a,b). Microbelts outcoupled efficiently the from the resonator. The PL collected from the microbelt tip displayed characteristic WGMs and the lasing threshold was weakly altered by the presence of the microbelt tail (14.2 and 15.9 μJcm^−2^ without and with microbelt) (Figure 15c,d). A different strategy exploited plasmon-photon coupling to create heterostructures composed of dye-doped polystyrene microspheres with 4–20 mm diameter with tangentially connected Ag nanowires of 170 nm diameter (Figure 15e,f) [141]. The surface plasmon polaritons (SPP) of the nanowires were efficiently launched by the optical modes confined in the microdisk. High photon-plasmon coupling efficiency between 1 to 3.5% were achieved as the result of WGM-SPP modes momentum match at the interface due to the tangential connection between the resonator and the nanowire. Interestingly, the mode distribution at the nanowire output was strongly dependent of the nanowire length, owing to the different dampings of modes located at low and high frequencies. This strategy can serve as a way to exploit the nanowires as frequency filters. 

Reducing the surface contact of the resonator with the substrate is an interesting way to favor light outcoupling and reduce optical losses induced by substrate leakage. Okada et al. demonstrated how the use of silver-coated substrate can favor the formation of edge on instead of face on crystals [140]. Enhanced optical confinement in this case enabled to reduce the lasing threshold by a four-fold. In this work, tuning of laser emission output was also achieved by developing WGM resonators based on rhombic co-crystals of blue and green emitting COPVs coupled by Förster resonance energy transfer (FRET) (Figure 15g–l). By tuning the concentration of the blue (donor) and green (acceptor) compound in the cocrystal, the FRET rate was tuned above or below the lasing rate of the donor, enabling for green or blue lasing. At a critical donor:acceptor concentration simultaneous blue and green lasing was observed (Figure 15m). 

## 5. Conclusions

Over the recent few decades, OMSCs have involved increasing attention as a promising field of knowledge and technology that involves physics, chemistry and materials science. The chemical versatility and flexible nature of the organic materials afford different nucleation, molecular packing and assembly between the organic compounds, allowing a control in the crystal growth. The intrinsic properties, molecular arrangement and the structure–property relationships of the OSMCs allow various crystal morphologies that can possess different electronic and optoelectronic properties. The current progress of OSMCs includes a wide range of techniques for achieving high quality OSMCs, without grain boundaries, defects, impurities and dislocations. The optimization of different methods for the crystal growth such ss solution-processing techniques (drop-casting, dip-coating, solvent exchange, solvent vapor diffusion) or vapor process crystals (PVT, microspacing sublimation) allows a control in the crystal morphologies. Solution-based methods are frequently utilized for organic molecules that exhibit large solubility in a wide range of organic solvents whereas PVT is the technique of choice for organic compounds that sublime without decomposition. Consequently, a huge effort in the optimization and exploration of single-crystals growth techniques have been developed to obtain high purity and diverse morphologies in the single crystals. These efforts have boosted the investigations into light emission and light amplification properties in the recent years. The important developments done in the field of OSMC lasers will pave the way to integrate them as basic elements in complex photonic circuits and optical logic gates. Such objectives will be enabled by exploiting lithography methods developed to control resonator size and shape. In these complex circuits, light coupling between the optical resonators and other optical elements will play a crucial role, so new strategies to boost light management in multicomponent structures will be of interest. OSMCs will also play an important role on the development of electrically-pumped lasers, owing to their outstanding charge transport properties. In this respect, a major goal will be the integration of the crystal on an LED-type sandwich structure with charge injecting electrodes without interfering with the lasing properties of the OSMC. Exploiting OSMC resonators for low-end optical pumped lasing (i.e., pumping with cost-efficient LED laser diode sources) is another promising milestone yet to be accomplished. This pumping scheme already demonstrated in conjugated-polymer-based DFBs should be on reach in OMSCs given their superior Q-factor. This result could be of interest for applications in solid state lighting as well as in the sensing field, where WGM and FP optical resonators are already employed as high sensitivity platforms to reveal changes in physical parameters or presence of chemical analytes through the shift of the resonator modes. In summary, OSMCs lasers will continue to attract attention in the next year due to their fascinating photonic and optoelectronic properties.

## Data Availability

No new data were created or analyzed in this study. Data sharing is not applicable to this article.

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
