# Peer review of "Organic Semiconductor Micro/Nanocrystals for Laser Applications"

_molecules, 2021, doi:10.3390/molecules26040958_

Round 1
Reviewer 1 Report
This review article is well-written and providing comprehensive overview regarding organic crystals and their laser applications. I belive that this article can be published as it is.
Author Response
We thank the referee for his positive comments.
Reviewer 2 Report
The paper describes important relationships among the molecular structures, packing modes, crystal morphologies, and optoelectronic properties of organic micro/nanocrystals especially focusing on laser application. Comprehensive reviewing on crystal growth methods, laser characteristics with a variety of cavity structures are of interests for researchers in this field, worth of being published in Molecules. Hopefully, another unique features of organic micromedia such as room-temperature polariton lasing, Raman lasing, superfluorescence could be also reviewed or commented in the conclusion section.
It is recommended to check the following miscellanea:
P3, line123: intermolecular vibrations à intramolecular vibrations?
P4, line159: contrast J-aggregates à contrast to J-aggregates,
P5, line188-189: This sentence should be deleted because afore-mentioned.
P16, line 592-593: Please check the sentence grammar.
P17, line 640: BP3T à BP2T? (Please describe full name of the molecules in not afore-mentioned).
Author Response
We thank the referee for the constructive comments. We have ammended the listed issues in the new version:
P3, line123: intermolecular vibrations à intramolecular vibrations? - Done
P4, line159: contrast J-aggregates à contrast to J-aggregates, - Done
P5, line188-189: This sentence should be deleted because afore-mentioned. - Done
P16, line 592-593: Please check the sentence grammar. : "A glance into the fluorescence image of a given microdisk (Figure 13b) indicates an alternate bright dark edge on the six lateral facets of the microdisk, thus confirming that a closed loop involves only three total internal reflections (3-WGM or 3D-WGM) at alternate faces, (Figure 13c). " was replaced by "The fluorescence image of a given microdisk (Figure 13b) depicts an alternate bright-dark edge, thus confirming that a closed loop involves only three total internal reflections (3-WGM or 3D-WGM) at alternate faces, (Figure 13c). "
P17, line 640: BP3T à BP2T? (Please describe full name of the molecules in not afore-mentioned). - The referee is right, BP3T was apprpriatly labelled as BP2T. We thank very much the referee for his careful revision.
Regarding the suggestion on citing other interesting optical properties like stimulated Raman, superfluorescence or polariton lasing, we feel that these are properties that remain a bit out of scope according to the view we intend to provide in this work. These are definitely interesting aspects which are worth to be treated separately. We thank in any case the referee for his suggestion.